# PERSPECTIVE

## The brain baroreflex – a hidden guardian of the cerebral circulation

Christabel Osei-Boateng [ID],
Jenna C. McCrone [ID]
and Michael M. Tymko [ID]

*Integrative Cerebrovascular and Environmental Physiology (ICEP) SB Laboratory, University of Guelph, Guelph, Ontario, Canada*

Email: mtymko@uoguelph.ca

Handling Editors: Kim Barrett & Philip Ainslie

The peer review history is available in the Supporting Information section of this article (https://doi.org/10.1113/JP288604#support-information-section).

Open up any medical textbook or research paper on cerebral blood flow control. It is a near guarantee that you will encounter a statement such as: 'the brain is a greedy organ, demanding approximately 20% of cardiac output despite making up a mere 2% of body weight!' The complex, but necessary, regulatory mechanism(s) governing this high level of cardiac output to the brain and its rigid infrastructure are mostly well understood. For reference, the highly cited 'Willie' brain outlines these mechanism(s) within a detailed review (Willie et al., 2014) – a must read for all young trainees. Yet, conspicuously absent from most discussions is a central, pressure-dependent mechanism that modulates arterial pressure to safeguard cerebral blood flow in the instance of prevailing intracranial pressure (ICP). Considering the potential devastating consequences of intracranial hypertension, the existence of such a safeguard seems intuitive.

Incredibly, the known impact of high ICP on cerebral function dates back centuries. To counteract this, one of the oldest medical interventions, which involves physically relieving pressure by removing a skull bone flap or drilling a burr hole (i.e. trephination) is sometimes implemented. The treatment approach of trephination dates back to the Neolithic period, with evidence found in skull collections from French Prunieres to Peruvian skulls as early as 10,000 BC (Moon & Hyun, 2017).

While the regulation of ICP through trephination is typically a last resort, a key discovery made by Dr Harvey William Cushing – the *Father of American Neurosurgery* – outlined that increased ICP may also have a role in the regulation of systemic blood pressure (Cushing, 1901). Using a canine model, Cushing meticulously documented what would later be known as the Cushing Reflex (see Fig. 1). His experimental protocol involved anaesthetizing dogs with morphine and inserting a needle into the femoral artery to measure systemic blood pressure and another into the cisterna magna to gauge cerebrospinal fluid pressure, which was then used as a surrogate for ICP. With a trephine opening in the skull, a glass window allowed direct observation of cerebral effects as ICP was altered by connecting a secondary skull opening to a bottle filled with warm saline. Raising or lowering the container adjusted ICP, much like modulating an intravenous bag's height to alter the flow of saline. Mercury tubes provided precise pressure readings of both arterial blood pressure and ICP. Cushing's 1901 paper summarized a simple yet profound central mechanism:

> As a result of these experiments, a simple and definite law may be established, namely, that an increase of intracranial tension occasions a rise of blood pressure which tends to find a level slightly above that of the pressure exerted against the medulla. It is thus seen that there exists a regulatory mechanism on the part of the vasomotor centre which, with great accuracy, enables the blood pressure to remain at a point just sufficient to prevent the persistence of an anaemic condition of the bulb, demonstrating that the rise is a conservative act and not one such as is consequent upon a mere reflex sensory irritation.

Cushing's discovery inspired further research in animal models, with subsequent studies by different research groups reaffirming his original findings. However, the question of whether this central mechanism existed in humans still remained unclear. Notably, with only an estimated 10–15% of animal research being

transferable to humans, it became necessary to design a human-based approach to confirm Cushing's century-old findings. With that, I am sure that most people reading this can appreciate that invasive measurements of ICP and mean arterial pressure (MAP) during manipulations in ICP in a human model is quite the challenge! Despite this, in 2018 Schmidt et al. provided evidence – for the first time in humans – that ICP regulation of arterial blood pressure can, in fact, be measured in patients with normal pressure hydrocephalus (Schmidt et al., 2005, 2018).

This clever study is one of our laboratory's personal favourites, as it bridged the gap between animal and human models. In the human participants undergoing an artificial cerebrospinal fluid infusion challenge to assess normal pressure hydrocephalus, the authors found a linear relationship between rising ICP, blood pressure and muscle sympathetic nerve activity measured via microneurography. A landmark investigation to say the least!

Fast forward to 2025 and we arrive at the impressive work of Wittenberg & colleagues, 'On the regulation of arterial blood pressure by an intracranial baroreceptor mechanism', in this issue of *The Journal of Physiology*. It provides compelling new evidence supporting the existence of an intracranial baroreceptor mechanism through a series of five animal experiments. Conducted in both anaesthetized and conscious rats, their experiments revealed a strong linear relationship between intracranial and systemic blood pressure, demonstrating that increases in ICP trigger proportional, non-habituating sympathetic nerve and cardiovascular responses. The authors also found that this central mechanism regulates blood pressure even during small changes in ICP. For example, their data demonstrated an ~10 mmHg decrease in MAP in response to acute craniotomy, associated with an ICP decrease from ~6 mmHg to 0 mmHg.

Interestingly, the cardiovascular responses to increases in ICP consistently outlast the duration of stimuli. The underlying mechanism(s), however, remains unclear, but the lasting responses are consistent with the hypothesis that astrocytes act as intracranial baroreceptors due to their mechanosensory properties. This

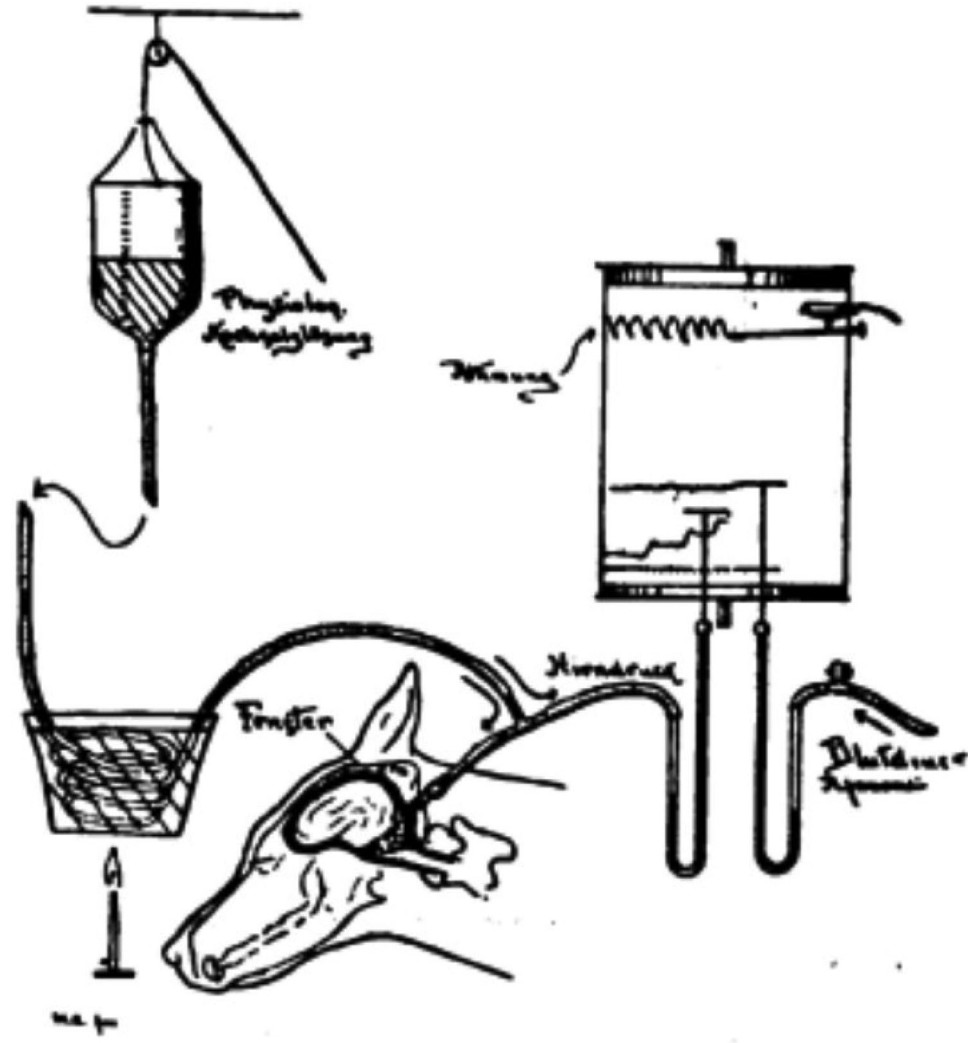

**Figure 1. Depiction of the experimental set-up of Harvey Cushing's seminal paper of 1901**
Adapted from Cushing (1901).

mechanism appears capable of effectively re-setting the arterial baroreflex, ensuring blood pressure rises sufficiently to offset reductions in CPP.

Wittenberg's comprehensive body of work provides further evidence for the presence of an intracranial baroreceptor that serves as a hidden guardian of cerebral circulation, ensuring that the constant demand for oxygen and nutrients is met under all circumstances. However, many questions remain, such as: What is the role of the brain baroreflex (if any) in the development of hypertension? How does the cerebral baroreflex function during spaceflight, or under other conditions, when 24-h ICP is chronically elevated? And finally, if the cerebral baroreflex increases arterial blood pressure by increasing sympathetic nerve activity, how does it function when sympathetic nerve activity is chronically heightened, such as at high altitude or in heart failure patients?

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

## Additional information

### Competing interests

The authors declare they have no competing interests.

### Author contributions

All others were involved in conception or design of the work and drafting the work or revising it critically for important intellectual content. All authors have read and approved the final version of this manuscript and agree to be accountable for all aspects of the work in ensuring that questions related to the accuracy or integrity of any part of the work are appropriately investigated and resolved. All persons designated as authors qualify for authorship, and all those who qualify for authorship are listed.

### Funding

M.M.T. received funding from the Canadian Government – Natural Sciences and Engineering Research Council of Canada (NSERC), 401965.

### Keywords

brain baroreflex, cerebral blood flow regulation, cerebral perfusion pressure, intracranial pressure

### Supporting information

Additional supporting information can be found online in the Supporting Information section at the end of the HTML view of the article. Supporting information files available:

**Peer Review History**

