## [Peer Review History · The Journal of Physiology]

The Brain Baroreflex – A Hidden Guardian of the Cerebral Circulation

Christabel Osei-Boateng, Jenna McCrone, and Michael M Tymko
DOI: 10.1113/JP288604

Corresponding author(s): Michael Tymko (mtymko@uoguelph.ca)

Review Timeline:

Submission Date:	06-Feb-2025
Editorial Decision:	12-Feb-2025
Revision Received:	14-Feb-2025
Accepted:	19-Feb-2025

Senior Editor: Kim Barrett

Reviewing Editor: Philip Ainslie

Transaction Report:

Dear Dr Tymko,

Re: JP-P-2025-288604 "The Brain Baroreflex - A Hidden Guardian of the Cerebral Circulation" by Christabel Osei-Boateng, Jenna McCrone, and Michael M Tymko

Thank you for submitting your manuscript to The Journal of Physiology. It has been assessed by a Reviewing Editor and an expert referee and we are pleased to tell you that it is acceptable for publication following satisfactory revision.

Please address all the points raised and incorporate all requested revisions or explain in your Response to Referees why a change has not been made. We hope you will find the comments helpful and that you will be able to return your revised manuscript within 4weeks. If you require longer than this, please contact journal staff: jp@physoc.org.

REVISION CHECKLIST:

- 'Potential Cover Art' for consideration as the issue's cover image
- Appropriate Supporting Information (Video, audio or data set: see https://jp.msubmit.net/cgi-bin/main.plex?form_type=display_requirements#supporting_information)

form_type=display_requirements#supp).

We look forward to receiving your revised submission.

Yours sincerely,

Kim Barrett
Senior Editor
The Journal of Physiology

EDITOR COMMENTS

Reviewing Editor:

Thank you for submitting a well-done Perspectives. please see minor, but very useful, comments from the reviewer.

REFEREE COMMENTS

Referee #1:

This perspective article by Christabel Osei-Boateng, Jenna McCrone and Michael Tymko discusses the findings of the new study by Wittenberg and colleagues, "On the Regulation of Arterial Blood Pressure by an Intracranial Baroreceptor", in relation to historical and more recent data on this subject. The article is well written, and I only have a few minor comments for the authors to consider in their revision:

1. I think the first evidence in humans was reported by Eric Schmidt in 2005: Schmidt, E. A., Czosnyka, Z., Momjian, S., Czosnyka, M., Bech, R. A., & Pickard, J. D. (2005). Intracranial baroreflex yielding an early cushing response in human. *Acta Neurochirurgica. Supplement*, 95, 253-256. This was followed by a more detailed study published in 2018.
 2. Lines 104-105, "For example, their data demonstrated an ~10-mmHg decrease in MAP in response to an ICP change from ~6- to ~0-mmHg". Perhaps better to say: "For example, their data demonstrated an ~10 mmHg decrease in MAP in response to acute craniotomy, associated with an ICP decrease from ~6 mmHg to 0 mmHg.
 3. Lines 110-112, "This mechanism appears capable of overriding the traditional arterial baroreflex, ensuring blood pressure rises sufficiently to offset reductions in CPP". The data reported by Wittenberg and colleagues suggest that this is not entirely correct. There is no evidence that the arterial baroreflex is overridden or blocked; although baroreflex sensitivity is reduced, it continues to effectively regulate sympathetic activity at all levels of MAP that follow changes in ICP. These data support the hypothesis that, under these conditions, the arterial baroreflex is reset centrally (similarly to what occurs during exercise). Perhaps it would be better to say: "This mechanism appears capable of effectively re-setting the arterial baroreflex, ensuring blood pressure rises sufficiently to offset reductions in CPP"
 4. Lines 130-133, The correct citation is: Cushing H (1901) Concerning a definite regulatory mechanism of vasomotor centre which controls blood pressure during cerebral compression. *Bull Johns Hopkins Hosp* 12: 290-292.
-

END OF COMMENTS

Dear Editor and Reviewer,

Thank you for providing feedback on our perspective article. We agreed with the reviewers feedback and integrated each of their suggestions into the article. Their feedback positively impacted the submitted article.

Dear Dr Tymko,

Re: JP-P-2025-288604R1 "The Brain Baroreflex - A Hidden Guardian of the Cerebral Circulation" by Christabel Osei-Boateng, Jenna McCrone, and Michael M Tymko

We are pleased to tell you that your paper has been accepted for publication in The Journal of Physiology.

Yours sincerely,

Kim Barrett
Senior Editor
The Journal of Physiology

If you would like to receive our 'Research Roundup', a monthly newsletter highlighting the cutting-edge research published in The Physiological Society's family of journals (The Journal of Physiology, Experimental Physiology, Physiological Reports, The Journal of Nutritional Physiology, and The Journal of Precision Medicine: Health and Disease), please click this link, fill in your name and email address and select 'Research Roundup':

<https://www.physoc.org/journals-and-media/membernews>

- You can help your research get the attention it deserves! Check out Wiley's free Promotion Guide for best-practice recommendations for promoting your work at: www.wileyauthors.com/eoo/guide. You can learn more about Wiley Editing Services which offers professional video, design, and writing services to create shareable video abstracts, infographics, conference posters, lay summaries, and research news stories for your research at: www.wileyauthors.com/eoo/promotion.

The Corresponding Author will receive an email from Wiley with details on how to register or log-in to Wiley Authors Services where you will be able to place an order
